# Allosteric signalling in the outer membrane translocation domain of PapC usher

Irene Farabella[1], Thieng Pham[2], Nadine S Henderson[3], Sebastian Geibel[1†],
Gilles Phan[1‡], David G Thanassi[3], Anne H Delcour[2], Gabriel Waksman[1],
Maya Topf[1*]

[1]Institute of Structural and Molecular Biology, Birkbeck College and University
College London, London, United Kingdom; [2]Department of Biology and
Biochemistry, University of Houston, Houston, United States; [3]Department of
Molecular Genetics and Microbiology, Center for Infectious Diseases, Stony Brook
University, Stony Brook, United States

**Abstract** PapC ushers are outer-membrane proteins enabling assembly and secretion of P pili in uropathogenic *E. coli*. Their translocation domain is a large β-barrel occluded by a plug domain, which is displaced to allow the translocation of pilus subunits across the membrane. Previous studies suggested that this gating mechanism is controlled by a β-hairpin and an α-helix. To investigate the role of these elements in allosteric signal communication, we developed a method combining evolutionary and molecular dynamics studies of the native translocation domain and mutants lacking the β-hairpin and/or the α-helix. Analysis of a hybrid residue interaction network suggests distinct regions (residue 'communities') within the translocation domain (especially around β12–β14) linking these elements, thereby modulating PapC gating. Antibiotic sensitivity and electrophysiology experiments on a set of alanine-substitution mutants confirmed functional roles for four of these communities. This study illuminates the gating mechanism of PapC ushers and its importance in maintaining outer-membrane permeability.

*For correspondence: m.topf@
cryst.bbk.ac.uk

Present address: †Institut für
Molekulare Infektionsbiologie,
University of Würzburg,
Würzburg, Germany; ‡Faculté
des Sciences, Université Paris
Descartes, Paris, France

Competing interests: The
authors declare that no
competing interests exist.

Reviewing editor: Volker
Dötsch, Goethe University,
Germany

## Introduction

Gram-negative pathogens commonly express a vast variety of complex surface organelles that are involved in different cellular processes. One of these organelles, known as pili (or fimbriae), forms a class of virulence factors involved in host cell adhesion and recognition, invasion, cell mobility, and biofilm formation. P pili from uropathogenic *Escherichia coli* are specifically required for the colonization of the human kidney epithelium, a critical event in the kidney infection process (pyelonephritis) (*Roberts et al., 1994*). P pili are assembled on the bacterial outer membrane (OM) via the chaperone/usher (CU) pathway (*Thanassi et al., 1998*), which is often used as a model system to elucidate the mechanism of pilus biogenesis (*Waksman and Hultgren, 2009*).

The biogenesis of pili via the CU pathway is a highly ordered process that comprises sequential steps. The chaperone protein (PapD) brings the pilins to the bacterial OM where they are assembled into a pilus at a transmembrane pore protein known as the usher (PapC). The usher (~800 residues) is composed of five domains (*Figure 1A*): a periplasmic N-terminal domain (NTD), an OM central translocation domain (TD) that comprises a translocation pore domain (TP), interrupted by a conserved Ig-like plug domain (PD), and two domains at the periplasmic C-terminal end (CTD1 and CTD2) (*Thanassi et al., 2002*; *Ng et al., 2004*; *Capitani et al., 2006*; *Phan et al., 2011*; *Geibel et al., 2013*). The structure of the apo TD (*Figure 1B,C*) consists of a 24-stranded kidney-shaped β-barrel where the PD is inserted into the loop connecting two β-strands (β6–β7), occluding the luminal volume of the pore (*Remaut et al., 2008*; *Huang et al., 2009*). In the activated form of

**eLife digest** *Escherichia coli* is a bacterium that commonly lives in the intestines of mammals, including humans, where it is usually harmless and can even be beneficial to its host. However, some types of *E. coli* produce hair-like filaments called P pili that allow the bacteria to attach to the human urinary tract and cause disease. To pass through the outer membrane of the *E. coli* cell, the filaments have to travel through a protein in the membrane called PapC usher.

The PapC usher protein—which is also involved in the assembly of the P pili filaments—contains a tube-like part called a β-barrel that is usually blocked by another part of the protein called the 'plug domain'. For the P pili to pass through the β-barrel, the plug domain has to move. This movement is controlled by two parts of the PapC protein, known as the α-helix and the β-hairpin, but it is not clear how.

To address this question, Farabella et al. made computer models of the normal PapC protein and versions that lacked the α-helix and/or the β-hairpin. Looking at these structural models and analyzing the evolution of PapC proteins helped to predict that certain regions of the β-barrel may be involved in controlling the movement of the plug domain, and this was then confirmed experimentally. Farabella et al. propose that these regions—together with the α-helix and β-hairpin—control the opening and closing of the β-barrel.

Further work is needed to investigate how other parts of the PapC protein are involved in P pili formation. These new insights could prove useful in the development of alternative treatments to fight bacterial infection.

another archetypal member of the usher family, FimD, the PD is located outside the pore lumen in the periplasm, next to the NTD (*Phan et al., 2011*; *Geibel et al., 2013*). In addition to the PD, there are two secondary structure elements that uniquely characterize the large β-barrel structures of the usher TD (*Figure 1B*). The first element is a *β-hairpin* that creates a large gap in the side of the β-barrel, a feature unprecedented in previously known OM β-barrel structures (*Remaut et al., 2008*). This element (located between strands β5 and β6 of the barrel, *Figure 1C*) folds into the barrel lumen and constrains the PD laterally inside the barrel pore. Mutants lacking the β-hairpin show an increased pore permeability suggesting that the β-hairpin has a role in maintaining the PD in a closed conformation (*Volkan et al., 2013*). The second element is an *α-helix* (located on the loop between β13 and β14, *Figure 1B*), which caps the β-hairpin from the extracellular side. Mutants lacking the α-helix, or in which the interface between the helix and the PD is disrupted, present a remarkable increase in pore permeability, comparable with that of the mutant lacking the PD, suggesting a role for the helix in maintaining the PD in a closed state (*Mapingire et al., 2009*; *Volkan et al., 2013*).

The mutant lacking both the β-hairpin and the α-helix is defective for pilus biogenesis (*Mapingire et al., 2009*). It has been observed in other OMP β-barrels that such secondary structure elements (e.g., an α-helix that protrudes inside the barrel or packs against the transmembrane strands) can use complex allosteric mechanisms to mediate their function (*Naveed et al., 2009*). These are often combinations of large conformational changes ('global motions') dictated by the overall architecture (including movement of secondary structure elements) and smaller changes ('local motions', such as the motion of recognition loops and side-chain fluctuations) (*Liu and Bahar, 2012*). Additionally, it has been shown that important residues in terms of evolution (highly-coevolved or conserved) could have a pivotal role in mediating such allosteric communications (*Suel et al., 2003*; *Tang et al., 2007*).

In this study, to understand the allosteric mechanism leading to the plug displacement in PapC and the involvement of the α-helix and β-hairpin, we used a hybrid computational approach and verified our results experimentally. By combining sequence conservation analysis, mutual information-based coevolution analysis, and all-atom molecular dynamics (AA-MD), we modelled the interaction network within the native PapC TD as well as within different mutants lacking the α-helix, β-hairpin, and both. This unique computational approach allowed us to identify residues that are likely to be involved in the transmission of the allosteric signal between the α-helix, β-hairpin elements and the

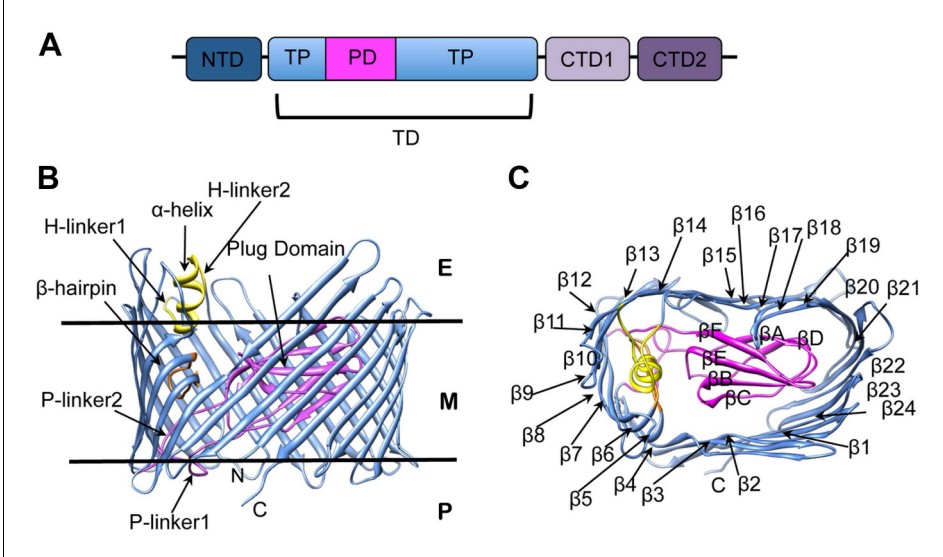

**Figure 1.** PapC usher organization and detail of its translocation domain. (**A**) A diagram of the domain organization of PapC usher. NTD (dark-blue) represents the N-terminal domain, CTD1 (light-violet) and CTD2 (dark-violet) represent the C-terminal domains; TD represents the translocation domain, comprising the TP (translocation pore, light-blue) and the PD (plug domain, magenta). (**B** and **C**): Ribbon representation of the starting model of the native translocation domain (TD) of PapC with the labels 'N' and 'C' indicating the N and C termini of the translocation channel. The β-barrel, PD (including the P-linkers), β-hairpin, and α-helix (including the H-linkers) are coloured blue, magenta, orange, and yellow, respectively. The outer membrane position is represented schematically with the labels 'E', 'M', and 'P' indicating the extracellular side, the membrane, and the periplasmic side, respectively. Side view of the TD (**B**) is shown with the α-helix, β-hairpin, H-linker1, H-linker2, P-linker1, P-linker2, and PD, labelled. Extracellular top view of the TD (**C**) is shown with the barrel β strands labelled β1 through β24 and with the PD strands labelled βA through βF. The figures were created with Chimera (*Pettersen et al., 2004*).

The following figure supplement is available for figure 1:

**Figure supplement 1.** MD simulations of the native PapC TD and its mutants.

plug. These residues were investigated by site-directed mutagenesis, functional studies, and planar lipid bilayer electrophysiology. The results confirmed the involvement of 4 of the 5 distinct communities of residues in modulating the usher's channel activity and gating, suggesting that they all participate in the allosteric mechanism controlling plug displacement.

## Results

To investigate if the β-hairpin or α-helix (or both) of the TD (residues 146–637 in the full length PapC) have a role in the allosteric communication leading to the displacement of the PD (residues 264–324), we performed four independent MD simulations, corresponding to the PapC TD model (sim1, *Table 1*) and three mutants embedded in a mixed lipid bilayer (*Table 1*): (i) where the region corresponding to the hairpin between β5 and β6 (residues 233–240) is deleted (sim2); (ii) where the α-helix between β13 and β14 (residues 447–460) is removed; and where both the regions were removed (sim4). The last 50 ns of simulation were considered for analysis, where the averaged root-mean-square deviation of $C_\alpha$ atoms ($C_\alpha$-RMSD) from the averaged structures stabilized around 2.00 ± 0.09 Å, 1.80 ± 0.09 Å, 1.86 ± 0.11 Å, and 2.03 ± 0.10 Å, for the *native* (sim1), *hairpin mutant* (sim2), *helix mutant* (sim3), and *helix-hairpin mutant* (sim4), respectively (*Figure 1—figure supplement 1*). This timescale, although limited for a full exploration of the structural changes induced by the mutations, was informative in revealing how local structural perturbations may affect allosteric changes leading to the plug displacement in PapC TD.

**Table 1.** Summary of the simulations.

| Simulation | Model systems | Length (ns) |
|---|---|---|
| Sim1 | Native PapC TD | 72 |
| Sim2 | Hairpin mutant | 70 |
| Sim3 | Helix mutant | 70 |
| Sim4 | Helix-hairpin mutant | 70 |

Descriptions of the items are: Simulation, the name of the simulation; Model systems, PapC TD model systems simulated; and Length, the length of the simulation.

## Non-covalent interaction network in the native PapC translocation domain and its perturbation in the absence of the β-hairpin, α-helix, or both

The changes in the non-covalent interactions (hydrogen bonds and salt bridges) between all residue pairs were analysed within the native TD by calculating their *non-covalent interaction score* (NCI score) (see 'Materials and methods'). A non-covalent residue–residue interaction network (RIN) comprising 492 nodes (residues) and 1350 edges (interactions) was then constructed as a weighted undirected graph for the native TD (*Figure 2*) and the three mutant systems (*Figure 2—figure supplement 1A–C*), with the weight for each edge given by the corresponding NCI score (*Table 2*). All four RINs have properties typical of small-world networks (*Atilgan et al., 2004*; *Haiyan and Jihua, 2009*; *Taylor, 2013*), with significant higher clustering coefficient compared to a corresponding random network and a higher mean short path length (*Table 2*). Within the constructed non-covalent native RIN, we identified 246 weak-to-strong interactions (connecting 362 nodes) with an NCI score of at least 0.3. Among these, 231 nodes connected by 133 edges showed an NCI score greater than 0.6 (i.e., strong interaction) of which 78 involve residues that are part of the barrel strands (58.6%).

Comparative analysis between the RINs of native and mutants systems revealed slight changes, suggesting a rearrangement in the interaction network. To better understand the mutation-induced changes in network components, we calculated the difference in non-covalent interaction score (ΔNCI score) between the native TD system and each of the mutant systems (the weakened interactions are shown in *Figure 2—figure supplement 2A–C*). This information was then added as a weighted undirected edge to the pre-existing native non-covalent RIN (the ΔNCI edges are shown in *Figure 2—figure supplement 2D*). Interestingly, 24% of the strong interactions in the native RIN were weakened relative to the RIN of the mutant lacking the β-hairpin, 22.6% relative to the mutant lacking the α-helix, and 23.3% relative to the mutant lacking both, suggesting that interactions between nodes that are not part of the deleted secondary structure elements were consistently weakened in the absence of these elements.

## Evolutionary analysis of PapC TD

We first extracted evolutionary information from a multiple sequence alignment of the PapC TD family. The patterns of conservation in the TD using Consurf (*Ashkenazy et al., 2010*) analysis suggested that the highly conserved residues (score 9) tend to be clustered in two specific regions of the usher (*Figure 3A*). The first cluster mapped onto the PD and the P-linkers (P-linker1 residues 248–263; P-linker2 residues 325–335) connecting it to the TP. The second cluster (which included the majority of the highly-conserved residues) mapped onto one side of the TP (strand β1–14 and β24). It includes residues: (i) near the periplasmic side of the β-barrel within β1–4 strands and β24 strand; (ii) on the extracellular side of the barrel (within β5–10); (iii) in the β-hairpin region (β-hairpin and β7–9); and (iv) in the area of β10–14 capped by the α-helix region, which comprises the α-helix and its linkers—H-linker1 (residues 445–450) and H-linker2 (461–468, respectively). Surface representation of the TD reveals a continuous patch of conserved residues facing the lipid bilayer, including β13, the extracellular half of β14 and the periplasmic half of β12 (*Figure 3B*). Intriguingly, this patch ('β13 conserved patch') reaches the full height of the pore from the α-helix region to a functionally important loop located between β12 and β13 strands (*Farabella, 2013*; *Volkan et al., 2013*).

In addition to investigating conservation, we performed an analysis to identify the coevolutionary relationships between residues in the structure. Using normalized mutual information (NMI) analysis (*Martin et al., 2005*) with a Z-score cut-off = 4 (see 'Materials and methods') to detect the intramolecular coevolved residues within PapC TD, a coevolutionary RIN containing 100 coevolved residues (nodes) and 357 connections (edges) was derived (*Figure 3D*). Mapping the network onto the PapC TD structure showed that many of the residues involved are also connected spatially and are

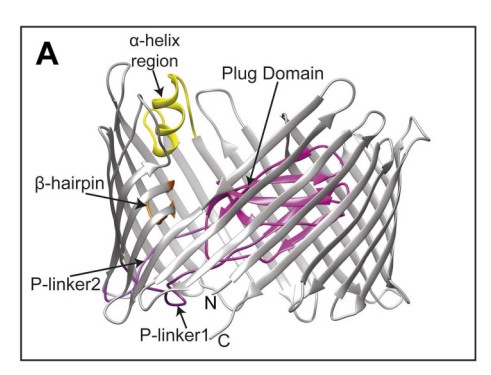

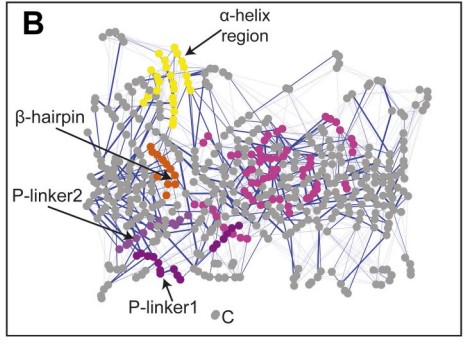

**Figure 2.** The native TD and its non-covalent interaction network (non-covalent RIN). (**A**) Ribbon representation of the starting model of the native translocation domain (TD) of PapC with the labels 'N' and 'C' indicating the N and C termini of the translocation channel. The β-barrel, PD, P-linker1, P-linker2, β-hairpin, and α-helix (including the H-linkers) are coloured grey, magenta, light purple, dark purple, orange, and yellow, respectively. The α-helix, β-hairpin, P-linker1, P-linker2, and PD are labelled. (**B**) Non-covalent RIN representation of the native translocation domain (TD) of PapC visualized with Cytoscape 2.8.2 (*Smoot et al., 2011*) based on RINalyzer plug-in analysis (*Doncheva et al., 2011*) (see *Figure 2—figure supplement 1* for the RINs of the TD mutants). The nodes (representing residues) are coloured by structural element as in (**A**) Edges (connecting two residues) are shown in blue, the edge width is proportional to its NCI score from lower to higher values.

The following figure supplements are available for figure 2:

**Figure supplement 1.** Non-covalent interaction network (non-covalent RIN) for PapC TD mutants.

**Figure supplement 2.** Combined RIN of the difference in non-covalent interaction score (ΔNCI score).

clustered in the same regions where the highly-conserved residues were found (P-linkers, the PD, and the barrel wall capped by the α-helix, in close proximity to the β-hairpin) (*Figure 3C,D*). The obtained coevolutionary RIN showed a significant clustering coefficient compared to a corresponding random network (of 0.493 vs 0.187, respectively) and a comparable mean short path length (3.15 vs 2.57, respectively) (*Daily et al., 2008*).

## Identifying allosteric 'hot spots' from a hybrid residue interaction network

We constructed one hybrid RIN in which the attributes for the nodes and edges are defined by the properties described above (non-covalent networks and evolutionary analysis, see 'Materials and methods'). Starting from the secondary structure elements (that uniquely characterise the barrel–the α-helix and β-hairpin) in this hybrid RIN, we used a multi-step procedure to reconstruct a pathway of communication between them (*Figure 4*).

This initial large sub-network is formed by 208 nodes (residues) connected by 456 NCI edges (in the native RIN). Applying the dynamic filter (independently) on the edges, based on the difference in non-covalent interaction score between the native TD and each of the mutants (ΔNCI > 0), revealed that in each case a large part of network has weakened interactions (hairpin mutant: 200 nodes, 438 NCI edges; helix mutant: 199 nodes, 437 NCI edges; helix-hairpin: 202 nodes, 443 NCI edges) (*Figure 4—figure supplement 1A*). The application of the evolutionary filter revealed that only a small part of the sub-network is made of evolutionary important residues (75 nodes connected by 104 native NCI edges). Combining the filters (*Figure 4—figure supplement 1B*) resulted in 69 nodes connected by 100 NCI edges (thus representing interacting residues in the native PapC network). The residues of this sub-network (14% of all residues in the TD) were considered 'hot spots' in the communication pathway of PapC TD ('hot spot' sub-network). Mapping them onto the structure revealed that they are located close together in a continuous area within PapC TD.

We analysed the community structure of the hot spot sub-network using the edge-betweenness clustering algorithm (*Girvan and Newman, 2002*; *Morris et al., 2010*). This analysis shows that the sub-network has a modular structure, with a modularity index of 0.73 (maximum value

**Table 2.** Summary of the residue–residue interaction networks (RINs) parameter.

| RIN | Full RIN | C | Cr | C/Cr | L | Lr | L/Lr |
|---|---|---|---|---|---|---|---|
| Native PapC TD | 1350 (492) | 0.384 | 0.012 | 32.00 | 6.67 | 3.78 | 1.76 |
| Hairpin mutant | 1196 (485) | 0.368 | 0.011 | 33.45 | 7.20 | 3.90 | 1.84 |
| Helix mutant | 1225 (476) | 0.362 | 0.011 | 32.90 | 6.67 | 3.90 | 1.71 |
| Helix-hairpin mutant | 854 (466) | 0.262 | 0.008 | 32.75 | 8.10 | 4.70 | 1.72 |

Descriptions of the items are: RIN, residue–residue interaction networks of the different model systems; Full RIN, number of edges in the RIN, in parenthesis the number of node; C, average clustering coefficient; Cr, average clustering coefficient for the random networks with the same size; C/Cr, average clustering coefficient ratio (as used in *Atilgan et al., 2004*); L, average shortest path length; Lr, average shortest path length for the random networks with the same size; L/Lr, average shortest path length ratio (as used in *Atilgan et al., 2004*).

of the modularity index is 1), which is typical of 3D-structure based RIN (*Newman and Girvan, 2004*; *Sethi et al., 2009*). Here, a total of 11 communities containing two or more residues were identified, from which only five communities are composed by more than five residues. For further analysis, we chose to consider only these five largest communities, which are located: between β7–9 and the P-linkers (C1); between the β-hairpin and the conserved region at the base of the α-helix (β12–14) (C2); between β12–β13_loop and the P-linker1 (C3); between the β-hairpin, P-linker2 and the PD (βE–F) (C4); and on the tip of the PD (βE–F loop and βA–B loop) (C5) (*Table 3* and *Figure 5*).

We selected a number of key residues from the communities (core hot spot residues), which link different elements within each community, for further experimental investigations (*Figure 5*). These were found in communities C1–C4: in C1, residues linking the P-linkers and the barrel wall that possibly help in maintaining the P-linkers in a closed configuration (P-linker1:D249, P-linker2:Y329, P-linker2:T331, β7:R337, and β8:S363); in C2, residues that bridge the base of the α-helix (the extracellular end of the β13 conserved patch) and β-hairpin (β-hairpin:R237, β12:S420, β13:R442, β13:S444); in C3, residues on the interface between P-linker1 and β12–β13_loop (P-linker1:Y260, β12–β13_loop:K427, β13:T437, β13:F438); and in C4, residues that are part of the interface with P-linker1 and the periplasmic end of the β13 conserved patch (P-linker2:A325, P-linker2:V327).

## Experimental analysis of residues in the hot spot sub-network

To test experimentally if the key hot spot residues identified above (linking elements within each community) contribute to allosteric signalling within PapC, we constructed a set of single alanine substitution mutations (*Table 4*). Each of the mutants was present at a similar level in the OM compared to the wild-type PapC usher, and the mutations did not affect the ability of the usher to form a stable β-barrel in the OM (data not shown). The functionality of the PapC substitution mutants was assessed by ability to assemble P pili on the bacterial surface. P pili bind to receptors on human red blood cells, and assembly of functional P pili was determined using a hemagglutination assay (HA). Seven (D249A, T331A, R442A, S444A, Y260A, K427A, T437A) of the 14 tested mutants exhibited greater than twofold defects in agglutination titers compared with wild-type PapC, with 4 of the mutants (D249A, R442A, Y260A, and K427A) exhibiting no agglutination activity (HA titer = 0) (*Table 4*). The defective mutants were in key residues from communities C1, C2, and C3, confirming roles for these communities in proper usher function.

We next used an antibiotic sensitivity assay to screen the PapC substitution mutants for effects on channel activity of the usher. The OM of Gram-negative bacteria has low permeability to detergents such as SDS and to antibiotics such as erythromycin and vancomycin, providing resistance to these molecules. In its resting state, the usher TP is gated closed by the PD, preserving integrity of the OM. Mutations that disrupt channel gating by the PD will result in the opening of the large TD channel, leading to increased sensitivity of the bacteria to antibiotics. Bacteria expressing five of the PapC substitution mutants (T331A, R237A, Y260A, F438A, and V327A) exhibited increased sensitivity to one or more of the tested molecules (*Table 4*). Y329A, R337A, S363A, and S420A did not appear to perturb the allosteric signalling within PapC, showing the same antibiotic sensitivity phenotype and ability to assemble pili of the native PapC (*Table 4*). However, the hemagglutination

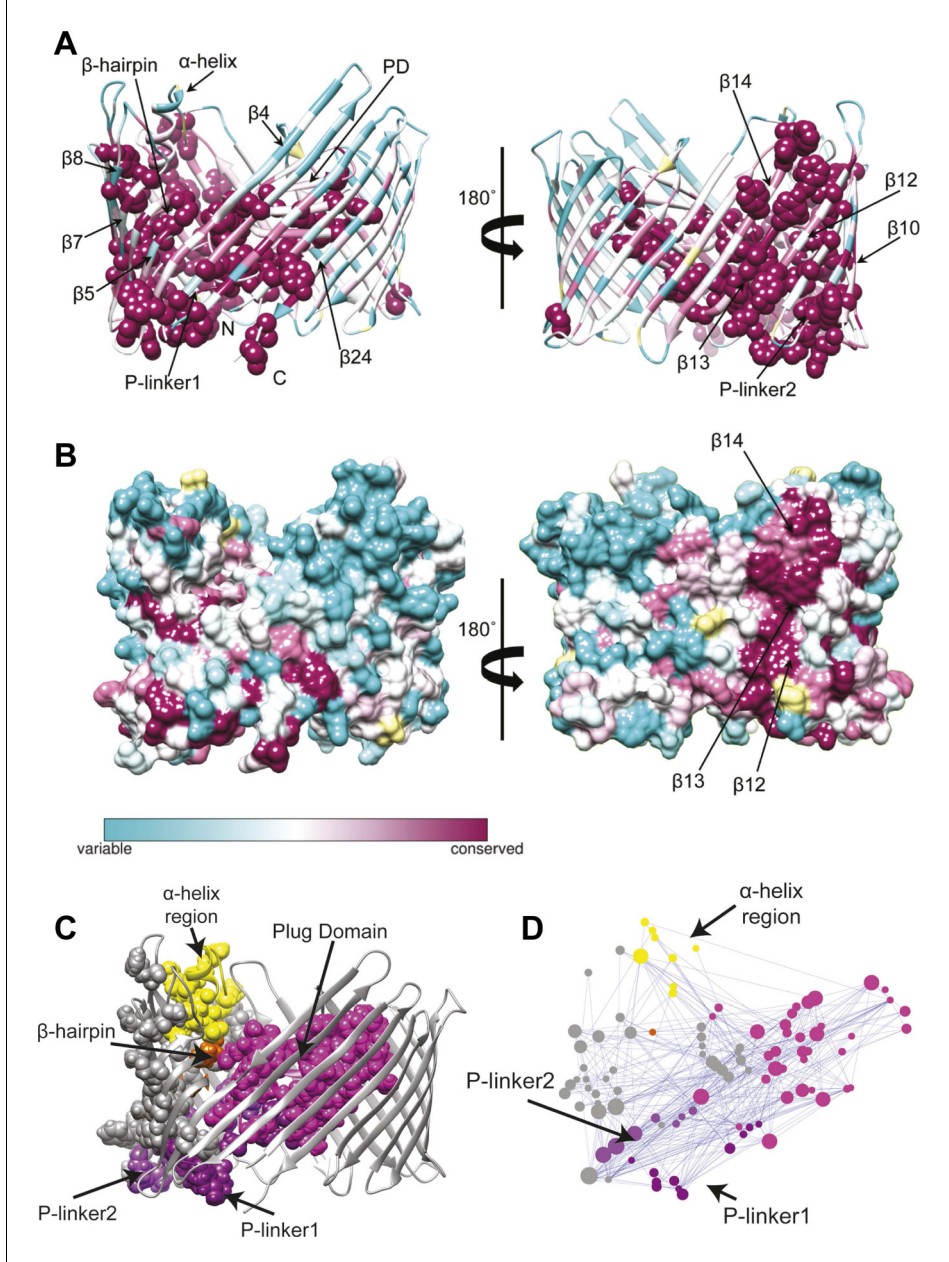

**Figure 3.** Evolutionary analysis of PapC TD. (**A–B**) Sequence conservation calculated with Consurf (**Ashkenazy et al., 2010**) and mapped onto the initial model of the native PapC TD (sim1, t = 0). Amino acid conservation scores are classified into nine levels. The colour scale for residue conservation goes from cyan (non-conserved: grade 1) to maroon (highly conserved: grade 9), unreliable positions are coloured light yellow. (**A**) Ribbon representation of the model with the highly conserved residues (grade9) shown as spheres and key elements labelled. (**B**) Molecular surface of the model with β12–β14 labelled. (**C–D**) Sequence co-evolution calculated with PyCogent (**Knight et al., 2007**; **Caporaso et al., 2008**). (**C**) The co-evolving residues are mapped onto the initial model of the native PapC TD (sim1, t = 0). (**D**) The co-evolution network as visualized with Cytoscape 2.8.2 Cytoscape 2.8.2 (**Smoot et al., 2011**) based on RINalyzer plug-in analysis (**Doncheva et al., 2011**). Edges (connecting two co-evolved residues) are shown in blue, and nodes (representing coevolved residues) are coloured by structural element. The PD, P-linker1, P-linker2, β-hairpin, and α-helix are indicated schematically and coloured as in **Figure 2**. The node size is proportional to its degree of connectivity.

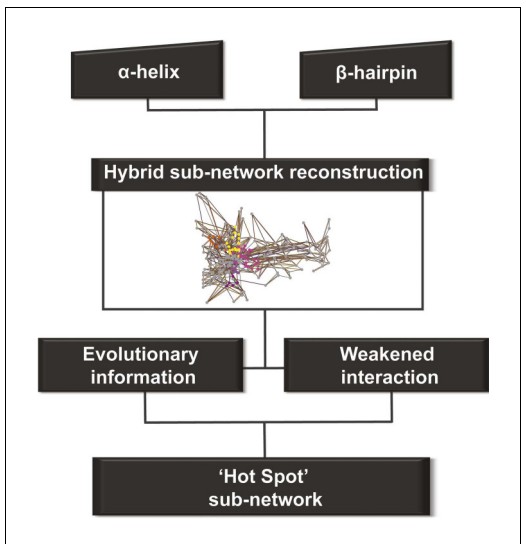

**Figure 4.** Detection of allosteric hot spots. A flowchart representing the multistep procedure used to identify allosteric hot spots. First, a sub-network of the protein hybrid RIN was generated starting from the α-helix and β-hairpin. Then, filters based on the evolutionary information and on the interactions analysis were applied (see *Figure 4—figure supplement 1*) resulting in a sub-network of 'hot spot' residues.

The following figure supplement is available for figure 4:

**Figure supplement 1.** Contribution of each filter in the detection of allosteric hot spots.

assay and antibiotic sensitivity assay are screening tools, and as such, they lack in sensitivity to pick up smaller changes in the ability to assemble pili or channel activity of the usher.

## Electrophysiological analysis of selected mutants

In total, 10 mutated PapC TDs were found to be affected either in their ability to trigger hemagglutination or in their permeability to SDS or antibiotics (T331A, D249A in C1; R237A, R442A, S444A in C2; Y260A, K427A, T437A, F438A in C3; and V327A in C4). We attempted to purify those mutants in view of examining their channel activity using planar lipid bilayer electrophysiology (which is a more sensitive assay). Unfortunately, only seven of these 10 mutants yielded protein stable enough (as wild-type) in detergent solutions to carry out the planned experiments (T331A in C1; R237A, S444A in C2; K427A, T437A, F438A in C3; and V327A in C4). During the OM extraction procedure, D249A, R442A, and Y260A were not stable enough due to the loss of the membrane bilayer environment and inability to maintain their native conformation in detergents. Insertion of PapC purified proteins (see 'Materials and methods') was promoted in planar lipid bilayers by clamping the membrane potential to −90 mV. As soon as channel activity was observed, the potential was briefly returned to zero and the chamber stirring stopped to minimize further insertions. 10-min long recordings of channel activity at + and −90 mV, and at + and −50 mV were performed.

The typical electrophysiological signature of the wild-type PapC usher is characterized by prolonged dwell times at a low current level, representing the closed state of the usher, and brief transitions of various current amplitudes. These transitions represent short-lived openings of various conductance, ranging from 50 to 600 pS ('transient-mixed' behaviour, TM) (*Figure 6A*). Although it is not possible to know exactly how many individual pores were inserted into the bilayer, the observed fluctuations of various sizes are taken to represent various conformational states of a single pore. As documented previously, the openings of the 'transient-mixed' behaviour appear rather small and may be due to the jiggling of the plug within the TP and/or the thermally induced mobility of various domains of the protein, such as the NTD and CTDs or loops (*Mapingire et al., 2009*). Occasionally, and more so at higher membrane potential, very large and sustained openings ('large-open' behaviour, LO) are observed in wild-type PapC usher (*Figure 6B*). These openings have a conductance of ~3–4 nS, which is similar to the monomeric conductance of the mutant lacking the PD

**Table 3.** Communities in the hot spot sub-network.

| Community | Residues |
|---|---|
| C1 | E247, D249, Y329, L330, T331, G334, Q335, R337, K339, E361, S363, W364, G365, L366, S371, L372 |
| C2 | R237, D402, S420, Y441, R442, F443, S444, K468, E469, M470, E475, W496 |
| C3 | Y260, Y425, S426, K427, T437, F438, A439 |
| C4 | S233, R303, G304, L306, V308, F320, T324, A325, V327 |
| C5 | E269, E312, N314, G315, R316, K318 |

Descriptions of the items are: Community, the name of the community; Residues, residues that are part of the community.

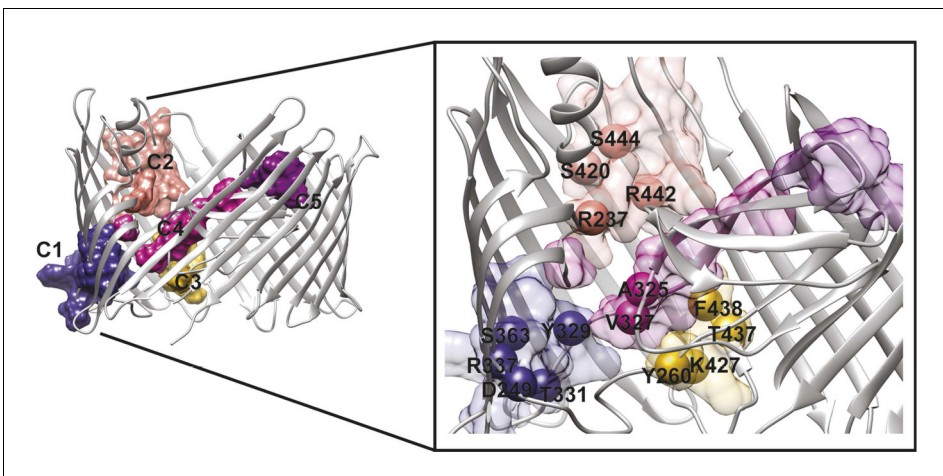

**Figure 5.** PapC TD communities. The communities of the hot spot sub-network are shown as surface by colours and indicated schematically (C1 to C5). The inset shows a close up of the identified core residues located in β7, β8, the P-linkers, the β-hairpin, the conserved region at the base of the α-helix, in the junction between β12–β 13_loop. The core residues are labelled in bold and numbered according to the X-ray structure of the apo PapC TD (PDB id: 2vqi).

(*Mapingire et al., 2009*) and are interpreted as representing a full displacement of the PD from a single monomer. Prolonged opening of intermediate conductance (0.5–1 nS) can also be observed and may represent partial PD displacement.

**Table 4.** Analysis of PapC substitution mutants.

| PapC | Community | HA titer | Antibiotic sensitivity | | |
|------|-----------|----------|------|------|------|
| | | | SDS | Erythromycin | Vancomycin |
| WT | | 64 | 15 | 6 | 6 |
| D249A | C1 | 0 | 15 | 6 | 6 |
| Y329A | C1 | 32 | 15 | 6 | 6 |
| T331A | C1 | 24 | 15 | 15 | 10 |
| R337A | C1 | 64 | 15 | 6 | 6 |
| S363A | C1 | 64 | 15 | 6 | 6 |
| R237A | C2 | 64 | 16 | 6 | 15 |
| S420A | C2 | 32 | 15 | 6 | 6 |
| R442A | C2 | 0 | 15 | 6 | 6 |
| S444A | C2 | 24 | 14 | 6 | 6 |
| Y260A | C3 | 0 | 15 | 14 | 6 |
| K427A | C3 | 0 | 14 | 6 | 6 |
| T437A | C3 | 24 | 14 | 6 | 6 |
| F438A | C3 | 32 | 14 | 12 | 6 |
| V327A | C4 | 64 | 20 | 14 | 16 |

Descriptions of the items are: PapC, the PapC construct tested; Community, the name of the community to which the mutated residue belongs; HA (hemagglutination assay) titer, the maximum fold dilution of bacteria able to agglutinate human red blood cells; Antibiotic sensitivity, the diameter of zone of inhibition (mm) around filter disc impregnated with SDS (750 μg), erythromycin (15 μg), or vancomycin (20 μg). The antibiotic sensitivity measurement includes the filter disc (6 mm diameter).

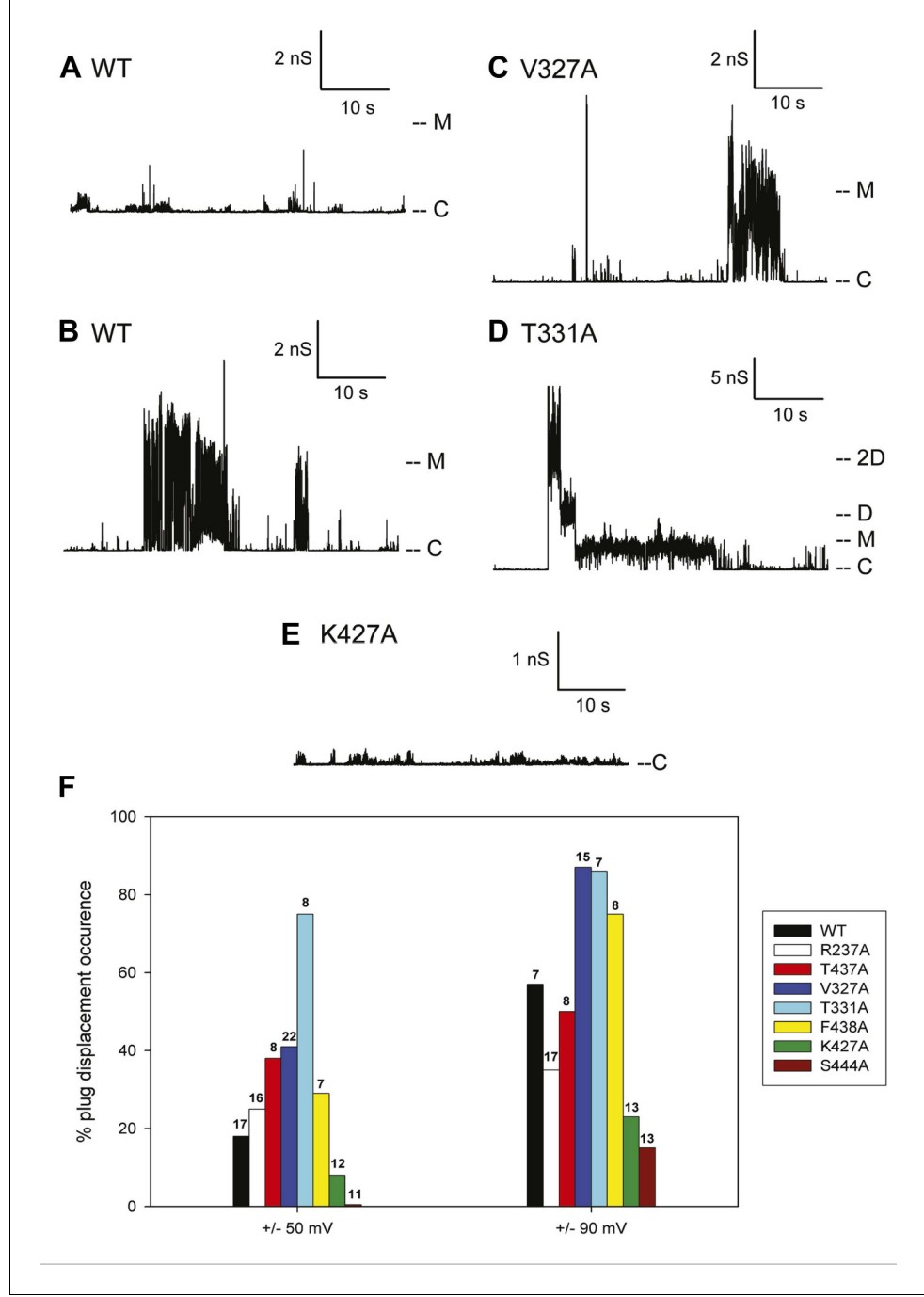

**Figure 6.** Kinetic signatures of channel activity in wildtype and mutant PapC ushers and frequency of PD displacement. Fifty-second segments of recordings obtained in planar lipid bilayers were selected to illustrate the behaviour of the different proteins. (**A**) Recording from the wild-type PapC usher showing the characteristic 'transient-mixed' behaviour. (**B**) Recording from the wild-type PapC usher showing an example of spontaneous large openings due to plug displacement. Note the large amount of current fluctuations during the openings, and the 'transient-mixed' behaviour in between such events. Examples of similar large openings (**C** and **D**) are shown for the V327A and T331A mutants, respectively. (**E**) A recording from the K427A shows that the channel barely displays any activity at this voltage. The voltage was +90 mV for all panels. The current level for the closed channels is marked as 'C', and openings are seen as upward deflections of the traces; current levels corresponding to fully open monomeric or dimeric forms are denoted by 'M' and 'D', respectively. Note that the traces are plotted as conductance, rather than current, vs time and the scale bars are given in nS. (**F**) The percent of sweeps

*Figure 6 continued on next page*

*Figure 6 continued*

displaying 'large-open behaviour' (LO) indicative of PD displacement is shown for WT and each mutant at the indicated voltages. The number of individual bilayers investigated in each case is given above the bars.

Because the electrophysiological behaviour of wild-type PapC usher is quite variable, and in attempt to quantify the propensity at spontaneous PD displacement, we have counted the number of 10-min long recordings (sweeps) that show 'large-open' behaviour, and we report the percent of such sweeps in various conditions. The frequency of observing these large openings in wild-type PapC usher is ~20% at ±50 mV, but increases to ~60% at ±90 mV. The application of a larger transmembrane voltage is likely to disrupt the interactions between key residues involved in keeping the PD in place, leading to a more frequent spontaneous displacement of the latter.

3 of the 7 analysed mutants, V327A, T331A, and F438A, showed an increased propensity at displaying large openings, relative to the wild-type PapC usher, as illustrated for V327A and T331A (*Figure 6C,D*). This was particularly true at ±90 mV where the percent of sweeps with large openings reaches values of 75–90% (*Figure 6F*). The T331A mutant was consistently more prone to open than WT and any other mutants, which led to the occasional simultaneous opening of several monomers (*Figure 6D*). Two of the seven mutants R237A and T437A still opened occasionally to the 3 nS level, but the frequency of sweeps with such events was slightly diminished relative to the wild-type PapC usher at ±90 mV (*Figure 6F*) suggesting that these mutants are likely to be insignificantly different from the wild-type PapC usher.

The K427A and S444A mutants showed a decreased frequency (or complete absence) of large openings. The K427A mutant almost never showed 'large-open' behaviour at ±50 and ±90 mV, indicating an extremely closed channel (*Figure 6E*, *Figure 6E,F*). The S444A mutant was even less prone to open, with 0% occurrence of PD displacement at ±50 mV in the 11 bilayers that we investigated (*Figure 6F*). However, increased activity with fast flickers and occasional more prolonged openings could be seen for both mutants if the membrane potential was switched to voltages in the ±100–150 mV range, indicating that the channels are present in the bilayer, but require higher voltages for activation.

## Discussion

PapC usher catalyses the translocation across the outer membrane of P pili, and its gating mechanism is important for bacterial homeostasis and for catalysis of pilus assembly. The TD of PapC is formed by the largest β-barrel pore known to be formed by a single chain. The PD occludes the pore in an inactive state and maintains the permeability of the channel. As previously documented, the native PapC channel is highly dynamic and is characterized by spontaneous short-lived openings of various conductance levels (*Mapingire et al., 2009*). Two distinct structural elements, the β-hairpin and the α-helix, play an important role in maintaining the PD in a closed conformation. In the absence of both elements the usher is defective for pilus biogenesis. Our analysis of the non-covalent interaction RIN in the native PapC TD shows that the interactions found between the TP and the PD are mostly weak, possibly to allow an easy release of the PD. This finding supports the idea that the highly dynamic behaviour of the native PapC channel is originated from the 'jiggling' of the PD within the TP (*Mapingire et al., 2009*). On the other hand, we find that the interactions between the TP and the P-linkers, between the TP and the β-hairpin, and between the TP and the α-helix region are mainly stable, supporting their role in maintaining the PD in a closed conformation.

Analysis of the mutation-induced perturbation of the non-covalent interaction RIN of the native PapC TD in the absence of the β-hairpin, α-helix, or both, shows that the interactions between nodes that are not part of the deleted secondary structure elements are consistently weakened (*Figure 2— figure supplement 1*). This feature suggests that the two elements are not independent and they are part of a complex allosteric process regulating the PD gating mechanism. It has been proposed that only a few residues play essential roles during allosteric processes and that perturbing the interactions between these residues can facilitate the population shift of the conformational ensembles (*Del Sol et al., 2009*; *Tsai et al., 2009*). Additionally, it has been shown that residues with a pivotal role in mediating such allosteric communications are also important in terms of evolution (both

highly coevolved residues and conserved residues) (*Suel et al., 2003*; *Ferguson et al., 2007*; *Tang et al., 2007*). Remarkably, we show here that PapC is characterised by an uneven distribution of the evolutionary important residues, clustered in the P-linkers, the PD, and the barrel wall capped by the α-helix, in close proximity to the β-hairpin. Another interesting finding is the presence of the 'β13 conserved patch' that reaches the full height of the pore from the α-helix region to the loop located between β12 and β13 strands (β12–β13_loop), which has been recently identified as important (*Farabella, 2013*; *Volkan et al., 2013*).

To detect the allosteric network, we implement a new method that integrates dynamic and evolutionary information in a hybrid RIN and then apply network analysis. This approach allows us to explore a large part of the protein, resulting in the detection of only 14% of all residues in the TD as potential candidates. It has been shown that detecting 'residue communities' in protein structure networks leads to the identification of key residues that are often part of a signal transduction pathway (*Bode et al., 2007*; *Del Sol et al., 2007*). The interconnection within and between the communities is pivotal for the flow of allosteric signalling. Residues in the same community are densely interconnected and have multiple routes to communicate with one another. However, the interconnections between communities involve only a few edges, which form the bottleneck for the flow of the signal in the network (*Bode et al., 2007*; *Del Sol et al., 2007*; *Sethi et al., 2009*). Here, all the identified communities (C1–C5, *Figure 5*) comprise residues from multiple elements (e.g., the β-hairpin, P-linker1, P-linker2, and distinct part of the TP) of PapC TD, except C5 that is composed only by PD residues. Mutation of key residues linking elements within each of the communities C1–C4 shows an altered antibiotic sensitivity phenotype, confirming a role of these communities in the pore-gating mechanism. Additionally, communities C1–C3 are required for proper usher function (as estimated by the hemagglutination assay), suggesting dependency of the pore gating function and pilus

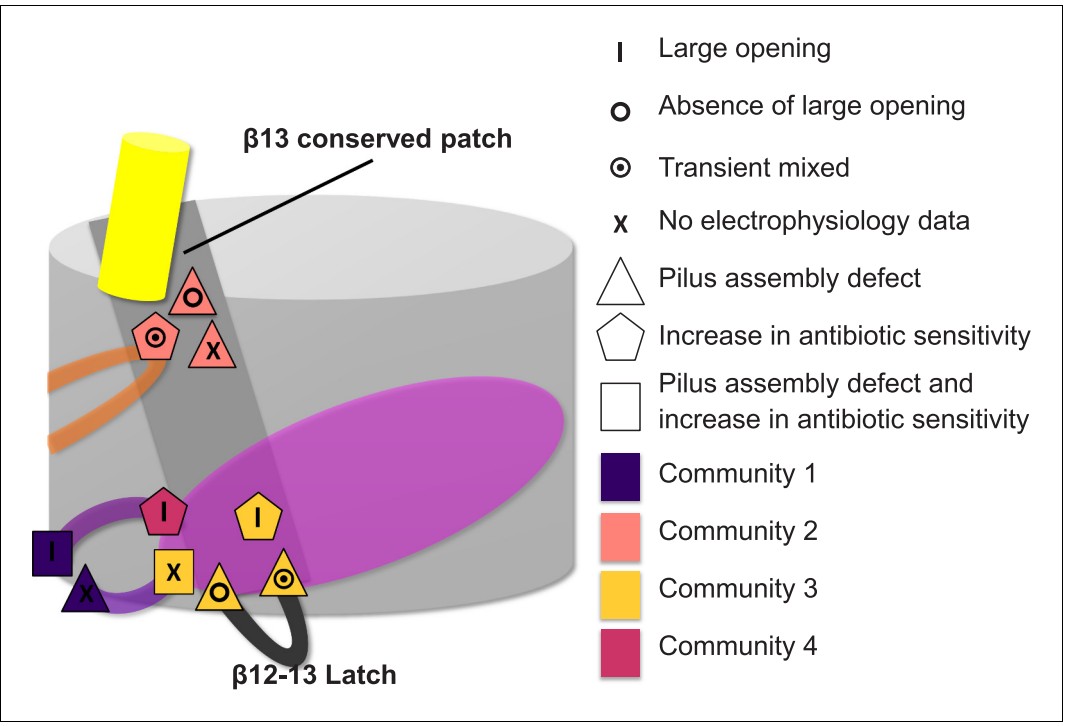

**Figure 7.** Residues involved in the allosteric signalling to control PapC gating. A schematic model summarizing the location of the detected hot spots involved in the gating mechanism. The β12–β13_loop (the 'latch') and the β13 conserved patch are coloured in dark grey and light grey respectively. The PD, P-linker1, P-linker2, β-hairpin, and α-helix are coloured as in *Figure 2*. Hot spot residues are colour-coded based on their communities (C1–C4, as in *Figure 5*). The different symbols indicate the mutant's electrophysiological behaviour ('X' where no data were available). Mutants that show a pilus assembly defect or an increased antibiotic sensitivity, or both, are represented by a triangle, pentagon and rectangle, respectively.

assembly (based on some of the mutations). For example, mutations in β12–β13_loop:K427 in C3 and β13:S444 in C2, which lead to a drastic decrease in the frequency of plug displacement (stabilizing the closed state, based on channel activity analysis), also show defective pilus assembly (possibly due to their deficiency in relocating the plug in a functional conformation).

Additionally, our study identifies two residues with opposite effects on plug displacement in the same community: C3: β12–β13_loop:K427, which is located on β12–β13_loop, and β13:F438, which is located at the periplasmic end of the β13 conserved patch (linking β12–β13_loop with the α-helix region). This observation indicates that β12–β13_loop has a pivotal role in the regulation of plug displacement by acting as a 'latch', as proposed previously (*Farabella, 2013*) and was also shown by mutagenesis studies (*Volkan et al., 2013*). Intriguingly, at the extracellular end of the β13 conserved patch there is another mutation (β13:S444) leading to a channel more reluctant to open, suggesting a regulatory role for the β13 conserved patch (*Figure 7*) in modulating the latch.

Interestingly, our community residues in some cases are found to have very different patterns of interactions in an alternative conformation. For example, the functionally important latch (β12–β13_loop) is shown to be in a different conformation in the open state (based on FimD:FimC:FimH structure [*Phan et al., 2011*]) as well as other residues in C3 and C4 communities. As a result, some residues that are identified by our method, such as V327, could in principle be selected also by visual inspection for further experimental investigation. V327 (located on P-linker2 in C4 at interface with P-linker1 and the barrel wall) is found to have a different interaction pattern in the open state of the usher and is also shown to promote plug displacement. However, more importantly, the method is able to predict residues in regions that would not attract our attention at all but may contain important information in respect to the allosteric pathway. (This aspect can become even more significant in the absence of an alternative conformation.) For example, we predict two such residues to be functionally important—T331 and S444. T331 is located on P-linker2 at the interface between P-linker2 and the barrel wall, which stays intact in the open conformation of the usher; S444 is located on β13 at the base of the α-helix, that is, on the barrel itself, and it lacks any direct contact with the plug domain. Surprisingly, and in support to our prediction, both residues are found to have an effect on plug displacement. Thus, the main strength of our method is that the knowledge of the structure of the protein (in one conformation only) and enough sequence information (to extract evolutionary information) are sufficient for the detection of functionally important residues that are pivotal for transferring the regulatory information within the protein.

## Conclusion

In this study, we provide a first deep insight into the allosteric regulation of the gating mechanism of the usher family. Using PapC TD as model system, we developed an integrative approach combining computational modelling, sequence conservation analysis, mutual information-based coevolution analysis and information from AA-MD simulations, to study the potential involvement of particular secondary structure elements (the α-helix and β-hairpin) in the allosteric communication. The construction of a hybrid interaction network and the use of network analysis allowed us to identify communities of residues within the TD that potentially mediate this process. Antibiotic sensitivity and electrophysiology experiments on a set of alanine-substitution mutants confirmed that residues located in the P-linkers, the β-hairpin, and β13 conserved patch (part of four communities) alter channel gating and that residues located in P-linker2, β12–β13_loop, and β13 conserved patch (both periplasmic end and extracellular end) are sensitive to plug displacement. Therefore, we suggest that the β13 conserved patch acts as a regulator of β12–β13_loop (the latch), mediating channel opening. Furthermore, our study shows how the integration of different computational approaches based on evolution, structure, and dynamics of proteins, into a hybrid network can unveil communication pathways within proteins. Such an integrative approach can guide the experimental investigation by pinpointing key candidates involved in the transmission of the allosteric signal.

## Materials and methods

### Systems modelling

We built four model systems based on the X-ray structure of the TD (residues 1–492) of PapC at 3.2 Å resolution (PDB ID: 2vqi [*Remaut et al., 2008*]). The starting model for the simulation of the native

TD (sim1, native) was generated by adding the missing loops to the X-ray structure using the *dope_-loopmodel* method (*Shen and Sali, 2006*) in MODELLER-9v7 (*Sali and Blundell, 1993*). Additionally, three mutant model systems were constructed based on the native model: a mutant lacking the β-hairpin (Δ233–240) (sim2, hairpin mutant), a mutant lacking the α-helix (Δ447–460) (sim3, helix mutant), and a mutant lacking both (Δ233–240 and Δ447–460) (sim4, helix-hairpin mutant).

Each of the systems was oriented with respect to the membrane normal (the Z axis by definition) using the database (*Lomize et al., 2006*). For the native model (sim1), a mixed lipid bilayer (POPE/POPG 3:1) was generated around the protein using the replacement method (*Jo et al., 2007*). To obtain a mixed lipid bilayer that reproduces an estimated surface area per lipid of 61.5 ± 0.2 Å$^2$ (*Murzyn et al., 2005*), we used the InflateGRO method (*Kandt et al., 2007*).

## Systems set up for MD simulations

All MD simulations were performed using Gromacs 4.0.5 (*Van Der Spoel et al., 2005*). TIP3P parameters were used for water molecules (*Jorgensen et al., 1983*), the OPLSA-AA force-field (*Kaminski et al., 2001*) was applied to the protein and ions, and the Berger force-field (*Berger et al., 1997*) to the lipids. All four systems were solvated in water and ions were added to neutralize the total charge (0.15 M NaCl), resulting in more than 75,000 atoms in total. Next, each system was energy-minimized using a steepest descent algorithm in the presence of different position restraints on the protein and the lipid bilayer head-groups, which were gradually removed. In the simulations of the mutant systems (sim2 to sim4), the protein models were embedded in the pre-equilibrated membrane obtained after 15 ns of unrestrained equilibration of the native TD (sim1).

## Equilibration procedure and production run

The assembled systems were equilibrated in a multistage process using periodic boundary conditions and a 2 fs time step. Short-range interactions were used with a cut-off of 1 nm. The PME algorithm (*Darden et al., 1993*) was used for long-range electrostatic interactions. All bonds were constrained using the LINCS algorithm (*Hess et al., 1997*; *Hess, 2008*). The first equilibration step was performed in the NVT ensemble, using a restraining force of 1000 kJ/(mol nm$^2$) for 0.1 ns on the protein and lipids. The V-rescale thermostat (*Bussi et al., 2007*) was employed to couple the temperature of the system to 310 K with a time constant of $t_T$ = 0.1 ps. All the following equilibration steps were performed in the NPT ensemble. During the next three steps, different parts of the system were restrained using a force constant of 1000 kJ/(mol nm$^2$): the protein and lipids, the protein atoms only, and the protein backbone atoms.

The resulting model of each system was then simulated without restraints. Constant temperature of 310 K was maintained using the Nose–Hoover thermostat (*Hoover, 1985*; *Nose, 1984*) with a time constant of $t_T$ = 0.1 ps. Using semi-isotropic coupling with a Parrinello–Rahman barostat (*Parrinello, 1981*), a constant pressure of 1 bar was applied with a coupling constant ($t_P$) of 1 ps and a compressibility 4.5 e$^{-5}$ bar$^{-1}$. Each unrestrained simulation was performed for ~70–72 ns. The last 50 ns of simulation was used for analysis.

## Non-covalent residue interaction network

Hydrogen bonds were defined using a cut-off of 30° for the acceptor–donor–hydrogen angle and a cut-off of 3.5 Å for the hydrogen-acceptor distance. The definition of salt-bridges was based on a 4 Å distance cut-off between any oxygen atoms of acidic residues and nitrogen atoms of basic residues. The non-covalent interaction score (NCI score) of the identified bonds was defined as the percentage of simulation time during which a bond occurs between two amino acids normalised by the number of bonds. Using the normalized score, a non-covalent residue interaction network (RIN) was built for each of the simulated systems as a weighted undirected graph, in which each node represents a residue and each edge is weighted by the normalized score. The difference in non-covalent interaction score (ΔNCI score) between the native TD system and each of the mutant systems was then calculated and added as a weighted undirected edge to the pre-existing non-covalent RIN.

## Sequence conservation analysis

The sequence corresponding to the X-ray structures of PapC TD (PDB id: 2vqi; Uniprot id: P07110) was used as input to psiBlast resulting in a set of unique related sequences from the non-redundant NCBI data set (*Altschul et al., 1997*). The E-value threshold was set as $10^{-3}$ and sequences with id >90% and <30% sequence identity were excluded. A structure-based multiple sequence alignment was calculated using Expresso (3DCoffee) (*Armougom et al., 2006*). Finally, an evolutionary conservation score was calculated for each residue an empirical Baysian inference method (*Mayrose et al., 2004*) as implemented in the ConSurf web server (*Ashkenazy et al., 2010*).

## Sequence coevolution analysis

To estimate the coevolution within the residues in the usher TD, we used the normalized mutual information (NMI) (*Martin et al., 2005*) over all position pairs in the multiple sequence alignment obtained as described above. NMI calculations were performed using PyCogent (*Knight et al., 2007*; *Caporaso et al., 2008*) and a Z-score was calculated for each residue pair based on the standard deviation from the mean NMI values. Only residue pairs that had Z-score > 4 were identified as coevolved pairs (*Gloor et al., 2005*; *Martin et al., 2005*).

Next, a coevolutionary RIN was built as a weighted undirected graph where each node represents a residue (as in the non-covalent RIN) and an edge connecting two nodes is the NMI score. The network was visualized and analszed in Cytoscape 2.8.2 (*Smoot et al., 2011*) using NetworkAnalyzer plug-in (*Assenov et al., 2008*) for calculating degrees of connectivity and RINalyzer plug-in (*Doncheva et al., 2011*) for mapping the network on the PapC structure.

## Hybrid RIN

To store the entire information, we combined the coevolutionary RIN, the non-covalent RIN, and the conservation analysis into one network. In this hybrid network each node represents each PapC TD residue and is associated with the corresponding conservation score. Two nodes can have multiple edges, each weighted according to the information it carries (NMI score, NCI score, or ΔNCI).

## Allosteric 'hot spots' sub-network reconstruction

Using the hybrid network and starting from the α-helix region (residues 445–468) and β-hairpin residues (residues 230–240), we generated a sub-network of first neighbours residues based only on the NCI score higher than 0.3 in the native RIN. This sub-network was expanded by again adding only neighbouring residues connected by NCI score higher than 0.3. The procedure was repeated until no new residues could be added to the sub-network. A first set was generated by filtering the sub-network based on the evolutionary information. The filtering was done by selecting nodes with a conservation score of 9 (i.e., highly conserved) or nodes that are connected by an NMI edge (i.e., coevolved significantly). A second set was generated by filtering out the nodes that are not connected by a weakened interaction (ΔNCI > 0) in each of the mutant systems. Intersecting the identified sets resulted in a 'hot spots' sub-network. The 'hot spots' sub-network was then decomposed into communities using the edge-betweenness clustering algorithm (GLay) as implemented in clusterMaker (*Girvan and Newman, 2002*; *Morris et al., 2010*). Cytoscape 2.8.2 (*Smoot et al., 2011*), RINalyzer plug-in (*Doncheva et al., 2011*), and Chimera (*Pettersen et al., 2004*) were used for mapping the network on the PapC structure.

## PapC substitution mutants

The PapC alanine substitution mutants were derived from plasmid pDG2 using the QuikChange Site-Directed Mutagenesis Kit (Stratagene/Agilent Technologies, Santa Clara, CA) and the primers listed in *Supplementary file 1*. Plasmid pDG2 encodes wild-type *papC* with a C-terminal, thrombin-cleavable His-tag (*Li et al., 2004*). All mutants were sequenced to verify the intended mutation.

## Expression and folding of the PapC substitution mutants in the outer membrane

Each of the PapC mutants was compared with wild-type PapC for expression levels and ability to fold into a stable β-barrel in the OM. OM isolation, analysis of usher protein levels, and the heat-

modifiable mobility assay for β-barrel stability were done as previously described (*Henderson et al., 2011*).

## Hemagglutination assay

HA assays were performed to test the ability of each of the PapC substitution mutants to assemble functional P pili on the bacterial surface. HA assays were performed by serial dilution in microtiter plates as previously described (*Henderson et al., 2011*). HA titers were determined visually as the highest fold dilution of bacteria still able to agglutinate human red blood cells. Each assay was performed in triplicate; each mutant was analszed twice and the values were averaged.

## Top soft agar assay for antibiotic sensitivity

Bacteria were grown in LB medium supplemented with 100 μg/ml ampicillin (Amp) to an $OD_{600}$ of 0.6 and then induced for PapC expression with 0.1% arabinose for 1 hr. Aliquots of 0.1 ml bacteria were added to 3 ml melted soft top agar (0.75% LB agar) cooled to 45°C and supplemented with 100 μg/ml Amp and 0.1% arabinose. The bacteria and melted agar were mixed well and poured on top of 1.5% solid LB agar plates containing 100 μg/ml Amp and 0.1% arabinose. Once the top agar solidified, sterile 6-mm filter discs were placed on top and 10 μl of the following antibiotics were added: 75 mg/ml SDS, 2 mg/ml vancomycin, or 1.5 mg/ml erythromycin. The diameter of the growth inhibition zone around the antibiotics, including the filter disc, was measured after overnight growth at 37°C. Each PapC mutant strain was tested twice and the values were averaged.

## Electrophysiological analysis of selected PapC substitution mutants

PapC mutants (R237A, V327A, T331A, K427A, T437A, F438A, and S444A) were purified according to published protocols (*Henderson and Thanassi, 2013*) and investigated by planar lipid bilayer electrophysiology. Planar bilayers were made from a preparation of L-α-phosphatidylcholine Type II-S from Sigma (also known as asolectin) according to the Montal and Mueller technique (*Montal and Mueller, 1972*) following a published protocol (*Mapingire et al., 2013*). Protein aliquots were diluted 1:1 in buffer T (1 M KCl, 5 mM Hepes, pH 7.2) containing either 1 or 2% *N*-Octyl-oligo-oxyethylene (octyl-POE, Axxora). Eight micrograms of protein from the diluted sample was added to the *cis* side of a planar lipid bilayer chamber containing ~1.5 ml of buffer T. Gentle stirring was applied to promote spontaneous insertions of the protein into the bilayer. Channel activity was monitored by measuring current under voltage clamp conditions using an Axopatch 1D amplifier with a CV4B headstage or an Axopatch 200B amplifier (Axon Instruments). The current was digitized (ITC-18; Instrutech) and stored on a PC computer using the Acquire software (Bruxton). 10-min long traces were sampled at 1.25 ms intervals and filtered at 500 Hz. Both chambers contained buffer T and Ag/AgCl electrodes with pellet. The *trans* side of the bilayer was set as ground. Insertions were typically performed at −90 mV. Data display and analysis were done with pCLAMP software (Axon Instruments).

## Acknowledgements

The authors are grateful to Dr David Houldershaw for computing support, and Drs Tsjerk Wassenaar and Paula Petrone for useful discussions. The authors acknowledge the use of the UCL Legion High Performance Computing facility, and associated services, in the completion of this work. The work was supported in part with MRC grant 018434 (to GW), a Wellcome Trust PhD Training Programme (086711/z/08/z) (to IF), MRC (G0600084) and BBSRC (BB/K01692X/1) (to MT) as well as National Institutes of Health grant GM062987 (to DGT).

## Additional information

### Funding

| Funder | Grant reference number | Author |
| --- | --- | --- |
| Wellcome Trust | PhD training programme 086711/z/08/z | Irene Farabella |

| Medical Research Council | G0600084 | Maya Topf |
| Biotechnology and Biological Sciences Research Council | BB/K01692X/1 | Maya Topf |
| National Institutes of Health | GM062987 | David G Thanassi |

The funders had no role in study design, data collection and interpretation, or the decision to submit the work for publication.

## Author contributions

IF, Conception and design, Acquisition of data, Analysis and interpretation of data, Drafting or revising the article; TP, Acquisition of data, Analysis and interpretation of data; NSH, Acquisition of data, Analysis and interpretation of data; SG, Acquisition of data, Drafting or revising the article; GP, Acquisition of data, Drafting or revising the article; DGT, Acquisition of data, Drafting or revising the article; AHD, Analysis and interpretation of data, Drafting or revising the article; GW, Conception and design, Analysis and interpretation of data, Drafting or revising the article; MT, Conception and design, Analysis and interpretation of data, Drafting or revising the article

## Additional files

### Supplementary files

• Supplementary file 1. Primers used in this study to generate PapC substitution mutations.

### Major dataset

The following previously published dataset was used:

| Author(s) | Year | Dataset title | Database, license, and accessibility information |
| --- | --- | --- | --- |
| Remaut H, Tang C, Henderson NS, Pinkner JS, Wang T, Hultgren SJ, Thanassi DG, Waksman G, Li H, | 2008 | Structure of the p pilus usher (PapC) translocation pore, http://www.pdb.org/pdb/explore/explore.do?structureId=2vqi | Publicly available at RCSB Protein Data Bank. |

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
