## [Decision Letter]

Thank you for sending your work entitled "Allosteric Signalling in the Outer Membrane Translocation Domain of PapC Usher" for consideration at *eLife*. Your article has been favorably evaluated by John Kuriyan (Senior editor) and Volker Dötsch (Reviewing editor), and two other reviewers (Nir Ben-Tal and Michael Wiener).

The Reviewing editor and the other reviewers discussed their comments before we reached this decision, and the Reviewing editor has assembled the following comments to help you prepare a revised submission.

The manuscript describes mechanistic studies of the outer membrane translocation domain of PapC Usher, a protein required for assembly and secretion of P pili in uropathogenic *E. coli*, which is involved in kidney infection. This is an interesting paper that utilizes a hybrid computational approach to identify amino acid residues involved in signaling/allosteric regulation. The use of multiple computational methods to select/filter those residues most likely to be significant adds confidence to their selection, and enabled the verification experiments to be better focused.

A few points should, however, be addressed:

1) While the three steps of RIN, evolutionary analysis, and interaction analysis are presented and well-described, the relative contributions of each to the final Hot Spot sub-network are not provided. For example, if evolutionary analysis were not included, how would this have affected the identified Hot Spot sub-network? Please include some information, like a Venn diagram, to illustrate how each step selected/filtered the results. Or, what if order of application of the steps were varied?

2) For the electrophysiology assay the authors wanted to purify all 10 mutants that had shown an effect in either the hemagglutination or in the permeability assay. Only 7 were stable enough for further analysis. This is to some extent a contradiction to the earlier statement that no difference in stability of the mutants relative to wild type protein has been seen in the OM of the bacteria. Is this only a detergent effect or can it be ruled out that these mutants are also less stable in the OM and therefore the results are influenced by differences in folding/stability?

3) The goal of this project was to use a combination of in silico tools to identify residues that play key roles in the allosteric regulation of the pore formation. One particularly important aspect that was noted by all reviewers is that this combination of multiple analytical methods probably decreases "false positives", i.e. residues that one or more methods selects as being functionally important, but another method (or two) filters out. Do you have identified residues that one or more methods select and that are removed by the remaining method(s)? A panel of three figures showing the residues that "light up" with each method would be useful.

In addition, are there residues located close to the allosteric residue communities in the structure that would have been selected by visual inspection of the structure as probably functionally important but that have not scored high in your combinatorial in silico approach? If there would be even experimental data showing that such "negative control" residues exist and can be predicted as functionally not important that would strongly support your method.

4) Along the same lines: 15 mutations were made but only 7 showed an effect in the hemagglutination assay and 10 if combined with the antibiotic resistance assay. Why do the other mutations not have the expected effect?

---

## [Author Response]

*1) While the three steps of RIN, evolutionary analysis, and interaction analysis are presented and well-described, the relative contributions of each to the final Hot Spot sub-network are not provided*. *For example, if evolutionary analysis were not included, how would this have affected the identified Hot Spot sub-network? Please include some information, like a Venn diagram, to illustrate how each step selected/filtered the results. Or, what if order of application of the steps were varied?*

We thank the reviewers for raising this issue, which is clearly very important but only partly addressed in the manuscript. We now added some new details about our method to clarify some of these issues.

First, in the Results we expanded the paragraph entitled “Identifying Allosteric ‘Hot Spots’ From a Hybrid Residue Interaction Network” that refers to the filtering issues (sentences starting “This initial large sub- network is formed by 208 nodes (residues) connected by 456 NCI edges (in the native RIN)”).

Second, we would like to highlight that the order of the filters does not affect the identified Hot Spot sub-network, which rises from the intersection of the different sets of networks resulting from the application of the above filters. To clarify this point we added a supplementary figure (Figure 4—figure supplement 1), showing a Venn diagram of each step involved in the method.

Additionally, we now updated the section entitled “Allosteric ‘hot spots’ sub-network reconstruction” to clarify that our filters are used in combination (sentences starting “Using the hybrid network and starting from the α-helix region (residues 445-468) and β-hairpin residues (residues 230-240) we generated a sub-network of first neighbours residues based only on the NCI score higher than 0.3 in the native RIN”).

Note that originally we wrongly stated in the Methods that we used a NCI score greater than 0.3 for the selection of the weakened interactions in the section above. Now this point is corrected.

Finally, we changed Figure 4 to clarify the issue of the order by which the filters are applied (i.e. the two filters are indicated in the same level).

*2) For the electrophysiology assay the authors wanted to purify all 10 mutants that had shown an effect in either the hemagglutination or in the permeability assay. Only 7 were stable enough for further analysis*. *This is to some extent a contradiction to the earlier statement that no difference in stability of the mutants relative to wild type protein has been seen in the OM of the bacteria. Is this only a detergent effect or can it be ruled out that these mutants are also less stable in the OM and therefore the results are influenced by differences in folding/stability?*

We appreciate the concerns of the reviewers. The overall expression levels and folding of the 10 examined PapC mutants in the outer membrane are, indeed, similar to wild-type PapC usher. All PapC variants used in the electrophysiology assay for single channel recording were equally stable during the OM extraction procedure as wild-type. However, some PapC mutants behave well in their physiological environment within the membrane, but are unstable once extracted out of the outer membrane for purification. This is the case in 3 of the 10 purified mutants. Indeed, during the OM extraction procedure, D249A, R442A and Y260A mutants were not stable enough to maintain their native conformation in detergents, likely due to the loss of the membrane bilayer environment.

In response to the suggestions, we added a sentence to the section entitled “Electrophysiological analysis of selected mutants” in the Results:

“During the OM extraction procedure, D249A, R442A, and Y260A were not stable enough due to the loss of the membrane bilayer environment and inability to maintain their native conformation in detergents.”

*3) The goal of this project was to use a combination of in silico tools to identify residues that play key roles in the allosteric regulation of the pore formation. One particularly important aspect that was noted by all reviewers is that this combination of multiple analytical methods probably decreases "false positives", i.e. residues that one or more methods selects as being functionally important, but another method (or two) filters out. Do you have identified residues that one or more methods select and that are removed by the remaining method(s)? A panel of three figures showing the residues that "light up" with each method would be useful*.

In addition, are there residues located close to the allosteric residue communities in the structure that would have been selected by visual inspection of the structure as probably functionally important but that have not scored high in your combinatorial in silico approach? If there would be even experimental data showing that such "negative control" residues exist and can be predicted as functionally not important that would strongly support your method.

We thank the reviewers for raising this interesting point.

The residues that “light up” as a result of the evolutionary information are shown in Figure 3. Additionally, to highlight the residues that light up as a result of the dynamic information in each of the mutants and their combination with the native PapC we have now added new supplemental figures: Figure 2—figure supplement 1 and Figure 2—figure supplement 2, showing each of the non-covalent RINs and their combination with the native PapC RIN, respectively.

As highlighted by the reviewers, the predictive power of our method rises from the combination of multiple analytical methods. These are based both on evolutionary data and on the information regarding the dynamics of the protein (without the need for prior knowledge on large confrontational changes it may undergo). As shown in the new supplementary figure (Figure 4—figure supplement 1) only 14% of all the residues in the TD were selected as a result of our integrative method, filtering out a large portion of the residues. Thus, the main strength of our method is the ability to explore a large part of the protein. As a result, the method is able to predict not only residues that could in principle be identified by visual inspection (especially if a structure of another conformation is available) but also residues in regions that would not attract our attention at all although they may contain important information in respect to the allosteric network. For example, V327, which was picked by our method and was shown experimentally to have a functional effect on plug displacement, could potentially have been selected for mutagenesis studies by visual inspection. This residue, which is located on P-linker2 (following a proline that is at the border of the mobile part of the linker) as part of the interface with P-linker1 and the barrel wall, has a different interaction pattern in the open state of the usher (based on FimD:FimC:FimH structure ([49] Jun 2;474(7349):49-53). On the other hand, the method predicted, for example, two residues to be functionally important, T331 and S444, and it is extremely unlikely that they would have been selected as candidates for mutagenesis studies by visual inspection only (out of the 494 residues comprising the TD). T331 is located on P-linker2 at the interface between P -linker2 and the barrel wall, which stays intact in the open conformation of the usher) S444 β13 is located on base of the α-helix, i.e. on the barrel itself, and it lacks any direct contact with the plug domain. However, both residues were shown to have an effect on plug displacement.

We have added a dedicated paragraph in a new Discussion section (sentences starting “Interestingly, our community residues in some cases are found to have very different patterns of interactions in an alternative conformation”).

*4) Along the same lines: 15 mutations were made but only 7 showed an effect in the hemagglutination assay and 10 if combined with the antibiotic resistance assay*. *Why do the other mutations not have the expected effect?*

Our method helps to guide the experimental investigation pointing towards regions of interest in the structure (“communities”). Along the line of the answer to question #3, it is clear that the method is useful to pinpointing key candidates while exploring a large part of the investigated protein without prior knowledge of conformational changes. Additionally one should keep in mind that the hemagglutination assay and antibiotic sensitivity assay (used in this study) are screening tools, and as such, lack in sensitivity to pick up smaller changes in the ability to assemble pili or channel activity of the usher. For this reason we further investigated the key candidate mutants using a more sensitive assay such as single channel recording.

We have now added a sentence in the Results section entitled ‘Experimental Analysis of Residues In The Hot Spot Sub-Network’ to clarify this point:

“Y329A, R337A, S363A and S420A did not appear to perturb the allosteric signalling within PapC, showing the same antibiotic sensitivity phenotype and ability to assemble pili of the native PapC (Table 4). However, the hemagglutination assay and antibiotic sensitivity assay are screening tools, and as such, they lack in sensitivity to pick up smaller changes in the ability to assemble pili or channel activity of the usher.”